# PVDF Hybrid Nanocomposites with Graphene and Carbon Nanotubes and Their Thermoresistive and Joule Heating Properties

**DOI:** 10.3390/nano14110901

**Published:** 2024-05-21

**Authors:** Stiliyana Stoyanova, Evgeni Ivanov, Lohitha R. Hegde, Antonia Georgopoulou, Frank Clemens, Fahmi Bedoui, Rumiana Kotsilkova

**Affiliations:** 1Open Laboratory on Experimental Micro and Nano Mechanics (OLEM), Institute of Mechanics, Bulgarian Academy of Sciences, Acad. G. Bonchev Str. Block 4, 1113 Sofia, Bulgaria or stiliyana.stoyanova@utc.fr (S.S.); ivanov_evgeni@imbm.bas.bg (E.I.); kotsilkova@imbm.bas.bg (R.K.); 2Centre de Recherche de Royallieu, Roberval Laboratory for Mechanics, CNRS, Université de Technologie de Compiègne, 60203 Compiègne, France; lohitha.hegde@utc.fr (L.R.H.); fahmi.bedoui@utc.fr (F.B.); 3Department of Functional Materials, Empa—Swiss Federal Laboratories for Materials Science and Technology, Überlandstrasse 129, 8600 Dübendorf, Switzerland; antonia.georgopoulou@empa.ch

**Keywords:** PVDF nanocomposites, graphene, carbon nanotubes, hybrid fillers, crystallinity, thermoresistivity, Joule heating

## Abstract

In recent years, conductive polymer nanocomposites have gained significant attention due to their promising thermoresistive and Joule heating properties across a range of versatile applications, such as heating elements, smart materials, and thermistors. This paper presents an investigation of semi-crystalline polyvinylidene fluoride (PVDF) nanocomposites with 6 wt.% carbon-based nanofillers, namely graphene nanoplatelets (GNPs), multi-walled carbon nanotubes (MWCNTs), and a combination of GNPs and MWCNTs (hybrid). The influence of the mono- and hybrid fillers on the crystalline structure was analyzed by X-ray diffraction (XRD) and differential scanning calorimetry (DSC). It was found that the nanocomposites had increased amorphous fraction compared to the neat PVDF. Furthermore, nanocomposites enhanced the β phase of the PVDF by up to 12% mainly due to the presence of MWCNTs. The resistive properties of the nanocompositions were weakly affected by the temperature in the analyzed temperature range of 25–100 °C; nevertheless, the hybrid filler composites were proven to be more sensitive than the monofiller ones. The Joule heating effect was observed when 8 and 10 V were applied, and the compositions reached a self-regulating effect at around 100–150 s. In general, the inclusion in PVDF of nanofillers such as GNPs and MWCNTs, and especially their hybrid combinations, may be successfully used for tuning the self-regulated Joule heating properties of the nanocomposites.

## 1. Introduction

Usually, nanocomposites possess improved and remarkable properties due to the addition of nanoparticles, which change the electrical, thermal, and mechanical properties of the polymer matrix [1,2]. The electrical conductivity is ameliorated due to the formation of electrical pathways, mainly caused by the electron tunneling of the embedded nanofillers [3]. The improvement in the electrical properties of the nanocomposite depends on the type of filler, size, morphology, state of dispersion, and the filler concentration within the polymer as well as the polymer matrix itself. Due to the electrical pathways being formed inside the nanocomposite, carbon nanofillers are used to develop smart self-sensing nanocomposite materials [4].

In the last few years, there has been a growing interest in the study of the thermoresistive effect (the change in electrical resistance due to a temperature change) and the Joule heating (resistive heating produced by the flow of electric current) of polymeric nanocomposites. These thermoresistive properties have attracted scientific attention due to their potential applications in electronics, energy harvesting, and sensor applications [5,6,7,8]. Thermoresistive polymer-based materials need conductive particles to evaluate the change in resistivity of the material. It is possible to perform thermoresistive property measurements of a polymer-based nanocomposite and to evaluate the Joule heating effect exactly because the nanocomposites are conductible. Polyvinylidene fluoride (PVDF) with good mechanical, thermal, and piezoelectric properties is used as a thermoplastic matrix. PVDF has a complex, semi-crystalline structure (≈50% crystallinity) composed of five different crystalline polymorphs: α, β, γ, δ, and ε phases. One of the specific properties of PVDF is its piezoelectricity, owing to the presence of the β phase, which has a polar dipole moment and thus increases the piezoelectricity. The incorporation of carbon nanoparticles and nanoscale filler hybrids into the PVDF matrix is an intriguing subject that has gained significant attention due to the enhanced thermal and electrical performance of the obtained nanocomposites [9,10,11]. The nanoparticles lead to an increase in crystallite formation because of their nucleating effect.

Carbon nanostructures play an excellent role as fillers for polymers in ameliorating these properties. A review of the literature findings led to the observation that MWCNTs are more researched than are GNPs regarding their Joule heating behavior. In a recent paper, researchers reported on the self-healing ability of recycled thermoplastic polyurethane (TPU) filled with MWCNTs due to the Joule effect [12]. An interesting approach has been proposed that involves examining the Joule heating of epoxy/MWCNT composites using both numerical and experimental investigations, with good agreement between the two [13]. Regarding the thermoresistive behavior, MWCNTs as fillers have been investigated using various polymer matrices, such as high-density polyethylene (HDPE) [14], polysulfone (PSF) [15], epoxy [16], and fiber–epoxy compositions [17], as well as copolymer combinations, for example, polypropylene random copolymer (PPR) [18], polyethylene glycol (PEG), and polyurethane (PU) [19]. XRD and DSC analyses were performed to obtain information for the MWCNT-doped PEG and PU blend, and the results led to the conclusion that the physical mixing of PU and PEG decreases the crystallinity of PEG, which results in improved thermoresistive behavior for the crosslinked nanocomposites [19]. On the other hand, graphene was analyzed for its self-regulating effect (an equilibrium of the generated temperature reached at applied voltage) [20] and embedded in poly(vinylidene fluoride-co-hexafluoropropylene) (PVDF-HFP) [21]. The self-regulating heating devices were prepared using crystalline polymer EVA mixed with amorphous polymer (PS or PVAc) [20]. Several carbon black (CB) hybrid compositions were also reported for improved positive temperature coefficient (PTC) characteristics and piezoresistive properties [22,23]. Some authors also reported the thermoresistive behavior of hybrid CNT-based nanocomposites [17]. They discussed the dispersion of the CNTs, presenting four different morphologies that were then linked to CNT concentration, CNT arrangement and rearrangement, fiber properties, interfacial interactions, thermal expansion, and polymer thermal transitions, identifying them as the key parameters that can influence bulk thermoresistive responses. Another study investigated the thermoresistive properties of the hybrid carbon nanostructure of polysulfone nanocomposites reinforced with multilayer graphene sheets (GSs) and multiwall carbon nanotubes (CNTs) at a total filler concentration of 1 wt.% [24]. The thermoresistive behavior of the composition of 50:50 GSs to CNTs had the highest sensitivity, which was due to the feeble electrical network being easily influenced by the temperature. Studies on the thermoresistive effects of PVDF-based nanocomposites have been conducted on inclusions such as inorganic particles and compounds, such as Ni particles [25], ionic liquid (IL) bis(1-butyl-3-methylimidazolium) tetrachloronickelate ([Bmim]_2_[NiCl_4_]) [26], and bismuth titanate (Bi_4_Ti_3_O_12_) [27]. However, most of the previous studies on such effects reported in the referred literature were performed with copolymers filled with metal-based nanoparticles, and there is little information on PVDF nanocomposites composed of carbon nanofillers.

This study investigated nanocomposites composed of PVDF homopolymer and carbon-based nanofillers—graphene nanoplatelets and multi-walled carbon nanotubes—at 6 wt.% filler content. At this amount, the nanofiller percolates in the prepared nanocomposites and, thus, conducts electrical charges. This paper focuses on the investigation of the thermoresistive effect and the Joule heating and of the effects of GNPs and MWCNTs and their hybrid combinations of PVDF-based nanocomposites. Furthermore, the influence of hybrid fillers at different GNP/MWCNT ratios on the crystal structure and properties was analyzed. The crystalline structure and morphology linked with the thermoresistive and Joule heating behavior were analyzed to reveal the influence of nanofiller hybridization on nanocomposite performances.

## 2. Materials and Methods

### 2.1. Materials

The polymer matrix of polyvinylidene fluoride (PVDF) was Kynar^®^ 721 (powder form) produced by Arkema (Philadelphia, PA, USA), which was homopolymer grade, with an MFI of 15 g/10 min (230 °C, 3.8 Kg), a melting point of 168 °C, and a glass transition temperature of −40 °C. For the carbon nanomaterials, the selected nanofillers were graphene nanoplatelets (SE1233-GNPs), supplied by The Sixth Element (Changzhou, China), and multi-walled carbon nanotubes (MWCNTs) from Nanocyl NC7000 (Sambreville, Belgium). Some of the main characteristics reported by the producers are shown in Table 1.

### 2.2. Preparation of Nanocomposites

This study used nanocomposites of PVDF filled with 6 wt.% GNPs and MWCNTs and their hybrid combinations, which were fabricated by the melt extrusion technique reported elsewhere [1,28]. Composites were prepared in the MackGraphe partner laboratory, Sao Paolo, Brazil, within the frame of the H2020-MSCA-RISE-Graphene 3D project. The melt extrusion was performed to ensure homogeneous filler distribution and prevent the formation of agglomerates in the nanocomposite [28]. The polymer and the fillers were dried at 80 °C for 4 h in a vacuum oven. The dried PVDF powder was wrapped with the appropriate amount of GNP or MWCNT powder in a ball mill for 2 h at 70 rpm. The resultant powder mixture was extruded in a twin-screw extruder Teach-Line ZK25T (COLLIN Lab & Pilot Solutions GmbH, Maitenbeth, Germany) at temperatures of 160–175 °C and a screw speed of 60 rpm. The temperatures used for the extrusion process in the range of the melting point of the PVDF were selected for the good integration of the nanoparticles inside the polymer matrix. The bi-filler hybrids of GNP, CNT, or PVDF with 6 wt.% total filler content as varying filler ratios (in Table 2) were prepared by mixing the two monofiller composites in appropriate proportions in a second extrusion run to ensure the better dispersion of the fillers, while the neat PVDF was processed in one extrusion run. Additionally, the powder of the neat PVDF marked as PVDF* in Table 2 was directly hot pressed.

### 2.3. Preparation of Test Samples

The test samples for this study were prepared by hot pressing the nanocomposite pellets obtained by the extrusion with a Carver 3850 hot press (Carver, Inc., Wabash, IN, USA) at a temperature of 230 °C and a pressure of 10 MPa to promote the β-phase formation [29]. This elevated temperature was also applied in order to obtain thin test samples with a thickness of around 650 μm, which were cooled down under the applied pressure. The samples were cut by the Nova Laser Cutting Machine (Thunder Laser USA, LLC., Quitman, TX, USA).

### 2.4. Experimental Methods

#### 2.4.1. X-ray Diffraction Analysis

The crystal structure was analyzed by wide-angle X-ray scattering (WAXS) with a Bruker D8 Advance X-ray diffractometer (Bruker Corporation, Billerica, MA, USA) at room temperature using Cu-Kα radiation (wavelength 0.1542 nm) at a voltage of 30 kV and a current of 40 mA. Intensities were measured in the range 5° ≤ 2θ ≥ 90° with a step interval of 0.02°. Similarly, the nanofillers in powder form were analyzed using WAXS.

#### 2.4.2. Differential Scanning Calorimetry

Differential scanning calorimetry (DSC) was performed to determine the degree of crystallinity and thermal transition temperatures using the DSC-Q20 (TA Instruments, New Castle, DE, USA). Samples (8–10 mg) were sealed in a pre-weighed alumina pan, and DSC experiments were performed in the range of 30 to 210 °C at 10 °C/min with heat–cool–heat cycles under a nitrogen atmosphere with a gas flow rate of 40 mL/min. The results were analyzed using TA Universal Analysis 2000 Software (TA Instruments, New Castle, DE, USA).

#### 2.4.3. Scanning Electron Microscopy

Scanning electron microscopy (SEM) was performed to visualize the morphology of PVDF-based nanocomposites. Samples were cut in liquid nitrogen and then gold-coated. SEM images of the cross-section were taken with the Tabletop SEM HIROX SH 4000 (Hirox Europe, Limonest, France) at different magnifications. The SEM images were taken at 15 kV accelerating voltage and 110 mA emission current conditions.

#### 2.4.4. Thermoresistive Characterization

The thermoresistive behavior and Joule heating measurements were executed at the experimental setup, shown in Figure 1a,b, according to the protocol established by the research institute Empa, Dübendorf, Switzerland [30]. The samples were laser-cut into rectangular shapes with 11 cm × 1.5 cm dimensions for the thermoresistive measurement and 3 cm × 0.7 cm for the Joule heating characterization.

The thermoresistive characterization shown in Figure 1a was performed as the initial resistance was measured by a Keithley 2450 source meter (Keithley Instruments, Solon, OH, USA) at a constant voltage of 5 V, while the sample was heated up on a heating plate Digital Hot from Thermo Fisher Scientific (Waltham, MA, USA), and the temperature was recorded by a digital thermocouple type K from Fluke Corporation (Everett, WA, USA), placed in the middle of the sample. 

The Joule heating was tested by measuring the generated heat and current in the samples at the different voltages applied, i.e., a constant voltage of 8 V and 10 V. The measurement setup (Figure 1b) included a DC power supply FI 1233 (Française d’In-strumentation, Quartier Europe Centrale, Sainte Savine, France) as a source of voltage. The current was measured as one of the electrodes was connected to an AC/DC current clamp E3N (Chauvin Arnoux Group, PARIS Cedex 18, France), which was linked to a Datalogger Graphtec GL220 (Dataq Instruments Inc., Akron, OH, USA). The temperature was detected by a thermocouple type T smeared with OMEGATHERM™ “201” High Thermal Conductivity Paste (Omega Engineering, Inc., Norwalk, CT, USA) and attached in the middle of the sample. The thermocouple was also linked to the datalogger, ensuring simultaneous recording of the temperature and the current with measuring rate each second.

## 3. Results and Discussion

### 3.1. Structural and Morphological Characterization

#### 3.1.1. X-ray Diffraction Analysis

As previously discussed, PVDF is a semi-crystalline polymer possessing different crystalline phases. The X-ray analysis revealed precise information about the existence of the α, β, and γ phases in the melt-extruded PVDF-based nanocomposites. The neat PVDF, GNP, and MWCNT nanofillers and 6 wt.% filled monofiller and hybrid filler compositions were analyzed by wide-angle X-ray scattering (WAXS), and the obtained results are shown in Figure 2a,b. (The XRD diffractograms revealed that the formation of the α and γ phases prevailed). The presented results were analyzed in the range between 15° and 30° 2θ, at which the β phase was also observed. The XRD analysis was performed for a limited range of the diffractogram where the characteristic peaks for the α, β, and γ phases were observed, and it was used to perform the qualitative analysis for the information of the phases in the PVDF and its nanocomposites. The α and γ phases diffracted in planes (100) at 17.7°, (020) at 18.4°, (110) at 20°, (021/022) at 26.6°, and (111) at 28° 2θ [31,32]. The only peak corresponding entirely to the alpha phase was reflected in (120) plane 25.7° 2θ as found in the PDF database (ICDD PDF#00-061-1403). The formation of the β peak was detected at 20.7° 2θ [33]. The peak of the β phase was the most clearly expressed in the composition of 6% CNT/PVDF. In the hybrid 4.5% GNP–1.5% CNT–PVDF composite and 3% GNP–3% CNT–PVDF composite, a peak at 18.1° 2θ was observed. This was a C60 peak according to the ICDD Database software, version PDF-5+2024 (ICDD PDF#04-013-1332), showing the effect of the interaction between the fillers, GNPs, and MWCNTs. 

The crystalline structure of GNPs and MWCNTs fillers was also investigated and is shown in the diffractograms in Figure 2b. The characteristic diffraction peaks of the carbon nanofillers with a reflection of (002) appeared at 25.6° 2θ [34]. The narrower peak of MWCNTs suggested a more crystalline structure than that of GNP. The characteristic peak of the nanofillers was not distinguishable in the XRD patterns of the PVDF-based nanocomposites, as seen in Figure 2a, due to the peak of the α phase at a similar 25.7° 2θ coming from the polymer. 

The ratio of crystalline to amorphous regions detected by the XRD was evaluated, and the phase content was calculated according to the following equations [33]:α + γ % = (A_α+γ_/A_α+γ_ + A_α_ + A_β_ + A_amorph._) × 100,(1)
α % = (A_α_/A_α+γ_ + A_α_ + A_β_ + A_amorph._) × 100,(2)
β % = (A_β_/A_α+γ_ + A_α_ + A_β_ + A_amorph._) × 100,(3)
Xc = (A_α,γ,β_/A_α+γ_ + A_α_ + A_β_ + A_amorph._) × 100,(4)
where α + γ % is the percentage of the α + γ crystalline peaks; α % is the percentage of crystallinity of the α peak; β % is the percentage of the β-phase content; Xc is the percentage of total crystallinity; and A_α+γ_, A_α_, A_β_, and A_α,γ,β_ are the area of the peaks for the α + γ, α, and β phases and their sum and A_amorph._—the area of the amorphous halo—respectively. 

Based on the approach described in [33], a peak deconvolution method was used to determine the area of the peaks required for the calculation of the crystalline phase fraction. Lorentzian functions fitted the crystalline peaks, and the amorphous halo was fitted by a Gaussian function. The deconvolution was carried out by fixing the baseline parameters for each sample, allowing for an unbiased comparison of the results. The initial positions of the crystalline peaks were fixed at known values, and the amorphous halo was positioned between 17–19° and 23–25° 2θ. The area and full width at half maximum of the peaks were recalculated during the convergent iterations. The peak deconvolution method was performed using the First Derivative Peak Finding Algorithm with Origin 2019b software (OriginLab Corporation, Northampton, MA, USA). 

The results of the fitting for the analyzed samples are presented in Figure 3a–f. The crystallinity content of the nanocomposites is given in Table 3. The fitting of the diffractograms and the following calculations revealed the processing effect of the extrusion and its role in the increase of the amorphous fraction. This was demonstrated by comparing each of the extruded compositions (Figure 3b–f) with the hot-pressed PVDF* powder (Figure 3a). The hot-pressed PVDF* powder had the highest crystalline fraction due to the least amorphous halo. The extruded PVDF had a decreased crystalline area in favor of the amorphous halo. The nanocomposites extruded in two runs had the highest amorphous area.

The area of the α + γ phases in the nanocomposites decreased compared to the neat PVDF, promoting the formation of the β phase. The increase in the β phase was from 0.2% for PVDF to 3.1% for 6% GNP–PVDF and 11.2% for 6% CNT–PVDF.

From the comparison of the nanofillers, it became clear that MWCNTs were a more effective nucleating agent than were GNPs, enhancing the percentage of β phase in the PVDF by over three times. The 3% GNP–3% CNT–PVDF nanocomposite had a similar amount of β phase as did the 6% CNT–PVDF composite. These results aligned with the conclusion based on the DSC results where the β phase was presented by a shoulder melting peak (Figure 4a). The melting peaks (T_m1_ and T_m2_) confirmed the presence of the β phase in the 3% GNP–3% CNT–PVDF composite and 6% CNT–PVDF composite. The wide melting peak of the 4.5% GNP–1.5% CNT–PVDF composite also indicated the β-phase content of 7.8% in this composition. The 6% GNP–PVDF composite had the least amount of 3.1% β phase visualized also by the sharp melting peak. The overall crystallinity of the nanocomposites determined by XRD analysis was lower than that of neat PVDF due to the larger amount of amorphous phase in the nanocomposites. 

#### 3.1.2. Differential Scanning Calorimetry

DSC thermograms were used to analyze the influence of the nanofillers and to provide a quantitative assessment of the degree of crystallinity of the nanocomposites. The results are given by the first heating cycle to correspond with the analyzed XRD measurement completed after the hot pressing of the sample. The peaks observed from the second heating cycle were almost identical (as seen in Table 4). The thermograms for the melting and melt crystallization temperatures of the PVDF-based nanocomposites are presented in Figure 4a,b.

Melting temperature gives additional information about the existence of α and β phases, as can be concluded from the observed peaks, respectively. The melting points of the α and β phases were found to be within the range of 167–172 °C according to the literature [29]. Concerning the influence of the nanofillers during the melting process as opposed to the neat PVDF, the difference found was in the intensity and the form of the peaks, but there was no discernible difference in the main melting peak’s (T_m1_) temperature (Table 4). 

The effect of carbon nanofillers on the crystallization behavior of PVDF was also proven by the calculations of the degree of crystallinity. According to the enthalpy of melting calculated by the data from TA Universal Analysis 2000 Software (TA Instruments) and the enthalpy of the purely crystalline PVDF in the literature [35], the degree of crystallinity is given by Equation (5):Xc = (ΔHm/wΔH0) × 100,(5)
where Xc is the percentage of total crystallinity, ΔHm is the enthalpy of melting calculated from the mass-normalized area of the T_m_ peak, ΔH0 = 104.7 J/g is the melting enthalpy of purely crystalline PVDF [35], and w is the portion of the polymer in the composite.

The narrow melting peaks of the neat PVDF and the 6% GNP–PVDF composite revealed the homogeneity of the crystal structure. Previous studies [36,37] indicated that GNPs have a weak effect on the melting behavior of the PVDF matrix. These studies discovered that agglomeration occurs at concentrations of 0.5 wt.% and above, which is why no visible effect of the GNPs’ influence was observed, as indicated in Figure 4a. On the other hand, the MWCNTs are well known to promote the formation of the β phase, as evident from the melting peaks and the appearance of a shoulder peak in DSC thermograms [38,39]. As shown in Figure 4a, there was a visible widening in the peak of the 4.5% GNP–1.5% CNT–PVDF composite; therefore, the presence of both α and β phases could be concluded. The absence of a shoulder peak could be attributed to the lower amount of MWCNTs compared to GNPs in this hybrid composition. At an equal ratio of GNP/MWCNT, the shoulder peak appeared at 174.8 °C for the 3% GNP–3% CNT–PVDF composite, while the monofiller composite of 6% CNT and PVDF showed the second, β-phase melting peak at a higher temperature of 175.7 °C, as seen in Table 4. The shoulder peaks (T_m2_) indicated the shifting of the α to β phase and different crystal regions, confirming the findings in [29].

The obtained results found no remarkable difference in the degree of crystallinity (Xc) or in the melt crystallization peak (T_c_) values of the neat PVDF and the composites with different types of fillers. 

#### 3.1.3. Scanning Electron Microscopy

The transformation in morphology of the nanocomposites affected by the presence of carbon nanofillers was observed by scanning electron microscopy. Figure 5a–e show the microstructure of the cryo-fractured surfaces of PVDF and nanocomposites at a low magnification of 1000× (first column) and a higher magnification of up to 20,000× (second column). 

As can be seen, the addition of carbon-based nanofillers considerably changed the microstructure of the PVDF. In Figure 5a, the neat PVDF sample has a smooth surface as opposed to the rough surface of the nanocomposites presented in Figure 5b–e. The microstructure of the cut surfaces of mono- and hybrid filler nanocomposites was very similar, except for the 3% GNP–3% CNT–PVDF composite (Figure 5e), which demonstrated a more loose structure. This may be associated with a specific arrangement of the GNP and MWCNT fillers formed at this equal filler ratio, as seen in Figure 5e. As seen in Figure 5(b2), for the composition of 6% CNT and PVDF, many circle-shaped inclusions were embedded in the structure, which indicates good affinity between the MWCNTs and PVDF and the coverage of the nanoparticle. For the composition of 6% GNP–PVDF and that of 4.5% GNP–1.5% CNT–PVDF, similar cohesiveness was found. For all the nanocomposites in Figure 5b–e, it was observed that the nanofiller particles on the cut surfaces were fully coated with the matrix polymer, meaning that the fracture path went through the polymer matrix and was not on the polymer–filler interfaces. Therefore, it may be assumed that there was good filler-polymer adhesion, which is associated with strong interfacial interactions in all three types of nanocomposite systems. Moreover, a good dispersion of nanofillers in the nanocomposites may be seen in Figure 5b–e at higher magnifications, which probably occurred due to the effect of processing, i.e., the primary wrapping of polymer with the nanoparticles via ball milling followed by extrusion.

### 3.2. Thermoresistive Characterization

#### 3.2.1. Thermoresistive Behavior

Thermoresistive behavior seeks to observe the changes in the resistivity of the materials with temperature changes caused by the external environment. Measurements were conducted in the range of 25 °C to 100 °C, i.e., below the heat deflection temperature (HDT), which for the used PVDF grade, is around 110 °C [40]. 

The measured resistance was transformed to resistivity—*ρ*—using the dimensions of the sample via Equation (6):*ρ* = R × A/L,(6)
where R is the initial resistance, A is the area of the cross-section of the sample, and L is the length between the electrodes. The results were obtained from the average sum of three measurements per composition.

The relation between the resistivity and the temperature of the analyzed samples is given in Figure 6. For the monofiller compositions of 6% GNP–PVDF and 6% CNT–PVDF, no changes in resistivity vs. temperature were observed; however, the resistivity of the GNP-based composite was five times lower than that with MWCNT. This resulted from the fact that the mean size of the GNPs (<50 μm) was a decade larger than the MWCNTs length (1.5 μm), and the surface area of GNPs (SSA = 400–600 m^2^/g) was twice as large as that of the MWCNTs (SSA = 250–300 m^2^/g). Therefore, the large-sized GNP particles formed a denser network in the PVDF matrix, leading to low resistivity compared to the MWCNTs, as well as low sensitivity to temperature. The regression lines in Figure 6 suggest that the hybrid compositions were more influenced by the temperature than were the monofiller ones. In general, the higher temperature led to the displacement of the hybrid combination of the nanoparticles in the matrix, which resulted in changes in the electrical pathways formed by the nanoparticles that, respectively, produced a resistivity increase. In the 3% GNP–3% CNT–PVDF hybrid composition, the temperature had a slight impact on the resistivity, as seen from the slope of the regression line, but the quantity of conductive GNP and MWCNT particles above the percolation threshold was presumably stable enough to remain slightly affected by the temperature in this region.

However, in the composition of 4.5% GNP and 1.5% CNT/PVDF, a strong increase in resistivity was observed at temperatures above 75 °C. The DSC curve did not show a thermal effect around 75 °C. We assume that the temperature influence was expressed in changes in the percolation network of the hybrid nanocomposites, which led to the disruption of the percolation paths between the GNPs and MWCNTs. The observed changes in the percolation network around 75 °C reflected the disruption of the percolation network due to the thermal expansion of the matrix, which occurred at mesoscale which could not be detected by the DSC measurement. A schematic illustration is presented in Figure 7. In addition, the percolation threshold of the MWCNTs (Nanocyl NC7000) was reported to be around 2 wt.% [41]. In the 4.5% GNP/1.5% CNT–PVDF composition, the content of the MWCNTs was just before the percolation occurred. Our measurement data showed a resistivity for the MWCTN–PVDF nanocomposites of 3.9 × 10^6^ Ω⋅m for 1.5% CNT–PVDF, 1.1 × 10^6^ Ω⋅m for 2% CNT–PVDF, and 2.0 Ω⋅m for 3% CNT–PVDF. Therefore, based on these results, the percolation threshold of the MWCNT–PVDF nanocomposites appears between 2 wt.% and 3 wt.% MWCNTs. Considering the very low MWCNT content in the hybrid composition, the thermal expansion could be enough to disrupt the conductive electrical paths and thus hinder the conduction of MWCNTs. Accordingly, at temperatures above 75 °C, a sharp increase in resistivity was observed, resulting in two regions (Figure 6): a slight increase before 75 °C, followed by a drastic rise of around 500 Ω⋅mm above that temperature. 

According to the datasheet in [42], the coefficient of linear thermal expansion (CLTE) of the homopolymer PVDF Kynar 700 series as a function of temperature is 14 × 10^−5^/°C at 23 °C, 18 × 10^−5^/°C at 70 °C, and 20 × 10^−5^/°C at 85 °C. As can be seen, higher temperatures (70–85 °C) cause a larger CLTE because of increased thermal expansion. This disturbed the 1.5 wt.% nanotubes’ weak percolated structure in the GNP–1.5% CNT–PVDF hybrid composite 4.5% and raised the resistivity above 75 °C. The degree of crystallinity in semi-crystalline polymers, such as PVDF, is directly correlated with thermal expansion. Researchers reported [43] that amorphous regions in semi-crystalline polymers tend to have higher thermal expansion than do crystalline regions. Thus, the lowest crystallinity of the 4.5% GNP–1.5% CNT–PVDF composite (Table 3), compared to the neat PVDF and the other nanocomposites, confirmed the highest sensitivity to thermal expansion of this hybrid composite, which may provoke the disruption of the weak percolated structure of MWCNTs around the percolation threshold, that is, 1–2 wt.% for the extrusion-processed polymer composites [28,41].

The term “temperature coefficient of resistivity” (TCR) is the calculation of a relative change in resistivity per degree of temperature change, as calculated by Equation (7) [44]. The coefficient could serve as an evaluation of the sensitivity of the resistivity to temperature.
TCR = (*ρ* − *ρ*0)/*ρ*0 × (T − T0),(7)
where *ρ* is the maximum resistivity at 100 °C, *ρ*0 is the resistivity at 30 °C, T0 is 30 °C, and T is the maximum temperature at 100 °C.

The calculated TCR is given in Figure 8. In general, the obtained values are very small, confirming the low sensitivity of the resistivity to temperature in the studied region. Nevertheless, according to the resistivity results, the highest sensitivity of 1.25 × 10^−2^ 1/°C was yielded by the 4.5% GNP–1.5% CNT–PVDF composite, followed by 9.34 × 10^−3^ 1/°C for the 3% GNP–3% CNT–PVDF composite, and 5.15 × 10^−3^ 1/°C for the 6% GNP–PVDF composite, while the lowest one of 3.37 × 10^−3^ 1/°C was yielded by the 6% CNT–PVDF composite.

#### 3.2.2. Heating Elements

The Joule heating effect (the passing electric current converted into heat) was investigated by applying 8 and 10 V for a period of 600 s. Figure 9a–d present the results given by the average sum of three measurements per composition for the measured temperature and current at 8 and 10 V. As seen in Figure 9a, when 8 V was applied, the results followed the trend set in the thermoresistive characterization; i.e., the monofillers were at the extremes, and the hybrid compositions were in between. The composition of 6% GNP and PVDF reached the highest temperature of 50.6 °C, as it has the lowest resistivity, and that of 6% CNT and PVDF reached the lowest temperature of 42.5 °C because it had the highest resistivity (shown in Figure 6). All the compositions displayed in Figure 9a reached a self-regulating regime after 150 s. The maximum temperature and current are given in Table 5. By comparing the current with the generated temperature (Figure 9a,b) we saw a difference in the composition of 4.5% GNP, 1.5% CNT, and PVDF that could be explained by the difference in the electrical and thermal paths, as the generated heat was influenced by the presence of conductive nanoparticles inside the polymer matrix. 

To observe the behavior of the samples at a higher generated temperature, 10 V was also applied (Figure 9c,d). The self-regulation of the temperature was reached faster at around 100 s for the 6% GNP–PVDF, 3% GNP–3% CNT–PVDF, and 6% CNT–PVDF composites. The composition of 4.5% GNP, 1.5% CNT, and PVDF showed exceptional behavior when reaching 85 °C, which is similar to the result shown in Figure 6. As in the previous test for the composition of 4.5% GNP, 1.5% CNT, and PVDF, the thermal expansion of the polymer disrupted the percolation in this composition. At a higher voltage, the range of the standard error increased.

The stabilization of the temperature over time proved the self-regulating effect due to the GNPs and the MWCNTs in the PVDF nanocomposites. Furthermore, this effect could be tuned by applying different voltages.

The heat could be calculated by Joule’s law, expressed by Equation (8):H = V × I × t,(8)
where V is the applied voltage, I is the measured current, and t is the time at which the current is measured.

For the evaluated time, the maximum values of the temperature, current, and generated heat are shown in Table 5. When comparing the first and second cases at 8 and 10 V for the monofiller compositions, we observed that the higher voltage led to about twice as much generated heat. The hybrid filler compositions had similar self-regulating properties at 8 V applied as opposed to the second case at 10 V applied. For the 4.5% GNP–1.5% CNT–PVDF composite, the generated heat increased three times, reaching the highest generated heat compared to all compositions.

## 4. Conclusions

PVDF-based nanocomposites reinforced with GNPs and MWCNTs and their hybrid combinations at 6 wt.% filler content prepared by melt extrusion were studied. The crystalline structure of nanocomposites was investigated by DSC and XRD. The nanoparticle inclusions increased the β-phase content compared to the neat PVDF. It was observed that MWCNTs promoted the formation of the β phase, acting as a better nucleating agent than GNPs. The SEM analysis indicated good adhesion between the polymer and the nanofillers, which strongly affected the properties. The thermoresistive analysis revealed a temperature-independent behavior in the analyzed region of 25–100 °C for the monofiller nanocomposites. The bi-filler hybrids demonstrated a higher temperature coefficient of resistance and therefore higher sensitivity to the temperature, which was pronounced mostly at a ratio of 4.5:1.5 GNP/CNT. Nevertheless, the composition of 4.5% GNP, 1.5% CNT, and PVDF cannot be proposed for positive temperature-coefficient-resistant (PTCR) applications based on these results, and further investigations are needed. The nanocomposites embedded with GNPs were more conductive than were the MWCNT-filled ones. The measurement of the generated heat by the Joule heating effect led to the conclusion that the GNP and MWCNT inclusions led to a self-regulating effect in both applied voltages. In general, via the selection of the carbon nanofillers and the right hybrid filler combinations, the heating temperature and self-regulation may be successfully tuned by the applied voltage to meet the application needs.

## Figures and Tables

**Figure 1 nanomaterials-14-00901-f001:**
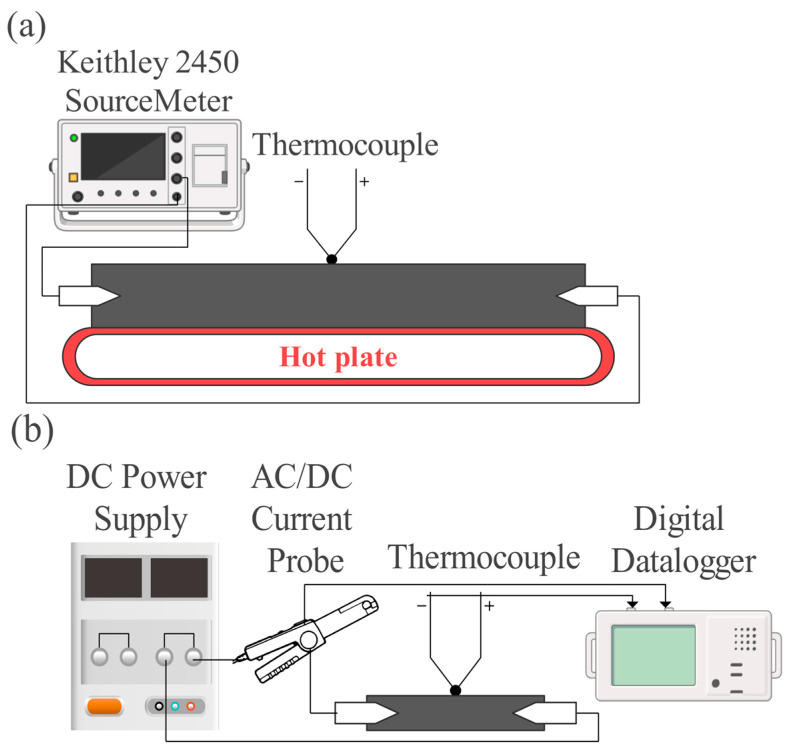
Experimental setup of (**a**) thermoresistive behavior and (**b**) Joule heating effect characterization.

**Figure 2 nanomaterials-14-00901-f002:**
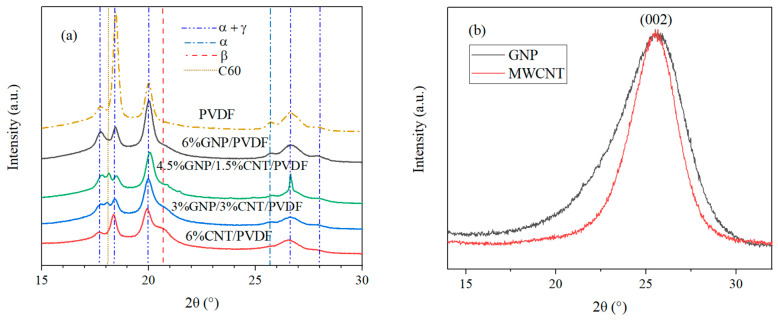
(**a**) XRD patterns of the neat PVDF and 6 wt.% nanocomposites; (**b**) diffractograms of the GNP and MWCNT powders.

**Figure 3 nanomaterials-14-00901-f003:**
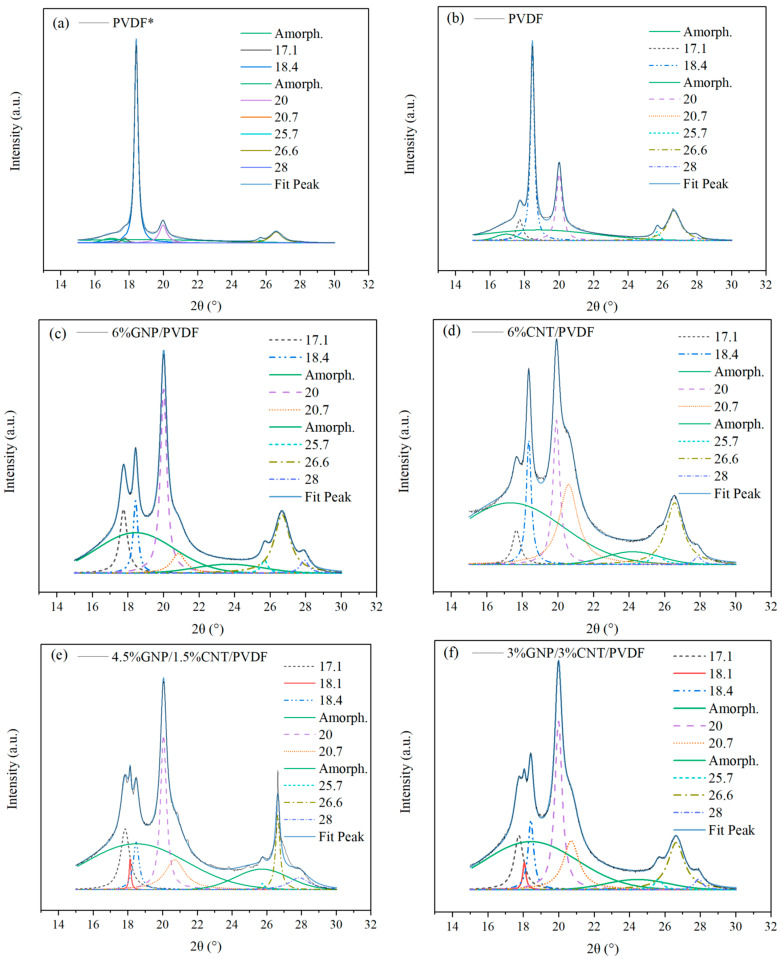
Peak deconvolution method of fitting. (**a**) Neat PVDF* directly hot-pressed powder. (**b**) Neat PVDF hot pressed after one extrusion run and the nanocomposites hot pressed after two extrusion runs: (**c**) 6% GNP–PVDF, (**d**) 6% GNP–PVDF, (**e**) 4.5% GNP–1.5% CNT–PVDF, and (**f**) 3% GNP–3% CNT–PVDF.

**Figure 4 nanomaterials-14-00901-f004:**
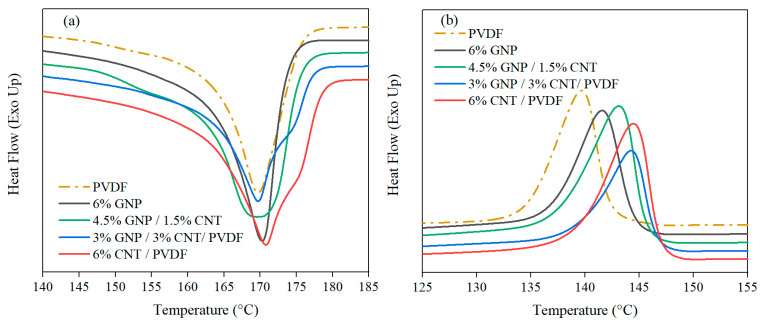
DSC thermograms of PVDF and its nanocomposites from (**a**) the first heating run with the melting peaks and (**b**) the cooling run with the melt crystallization peaks.

**Figure 5 nanomaterials-14-00901-f005:**
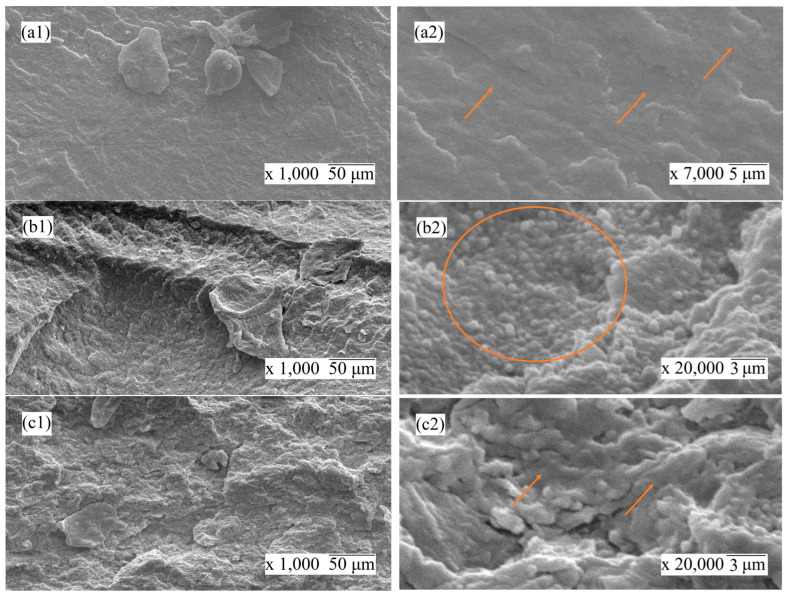
SEM images of the cryo-fractured surfaces of the compositions of (**a**) neat PVDF, (**b**) 6% CNT–PVDF, (**c**) 6% GNP–PVDF, (**d**) 4.5% GNP–1.5% CNT–PVDF, (**e**) and 3% GNP–3% CNT–PVDF. The first column is low magnification 1000×, and second column at a decade higher magnification of up to 20,000×. The noticeable specific areas of interest are marked with a circle and arrows.

**Figure 6 nanomaterials-14-00901-f006:**
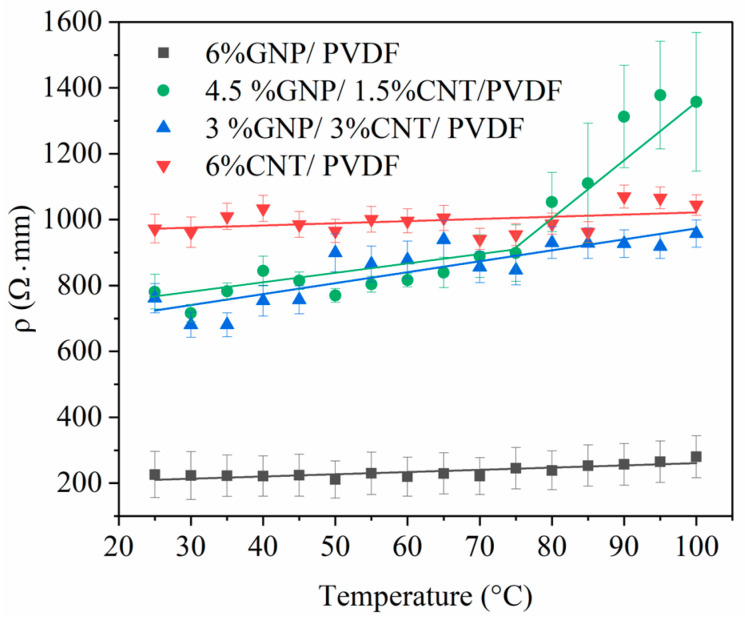
Resistivity vs. temperature of the 6 wt.% PVDF-based nanocomposites (experimental points) and the linear regression lines.

**Figure 7 nanomaterials-14-00901-f007:**
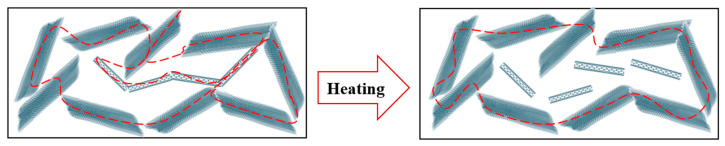
A proposed schematical representation of the disruption of the electrical paths between the GNPs and MWCNTs in the 4.5%GNP–1.5%CNT–PVDF composition due to thermal expansion.

**Figure 8 nanomaterials-14-00901-f008:**
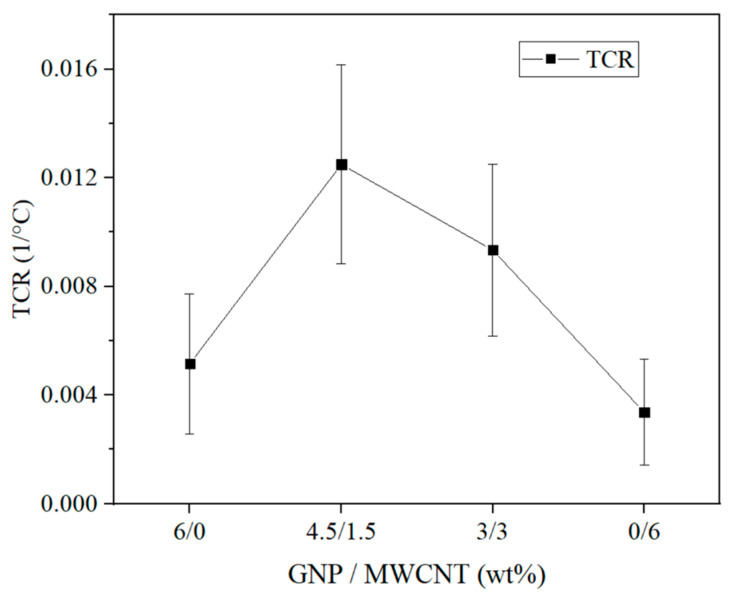
Temperature coefficient of resistivity (TCR) of the PVDF-based nanocomposites at 30 °C and the maximum temperature at 100 °C.

**Figure 9 nanomaterials-14-00901-f009:**
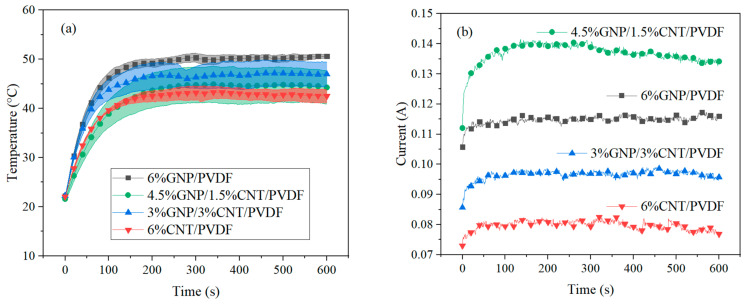
Joule heating analysis. Applied constant voltage of 8 V: (**a**) accumulated temperature and (**b**) current. Applied constant voltage of 10 V: (**c**) accumulated temperature and (**d**) current. The standard error is shown in (**a**,**c**) with the matching colors for the specified composite.

**Table 1 nanomaterials-14-00901-t001:** Characteristics of the carbon nanofillers, GNPs, and MWCNTs.

Filler	GNP	MWCNT
Trade Name	SE1233	NC7000
Purity, C wt.%	>97	90
Particle size, D_50_, μm	<50	-
Outer diameter, nm	-	9.5
Inner diameter, nm	-	5
Length, μm	-	1.5
SSA, m^2^/g	400–600	250–300
Shape	2D	1D
Volume Resistivity, Ω.cm	-	10^−4^

**Table 2 nanomaterials-14-00901-t002:** List of samples used in this study.

Sample	PVDF wt.%	GNP wt.%	MWCNT wt.%	GNP/MWCNT wt.%
PVDF*	100	-	-	-
PVDF	100	-	-	-
**Monofiller composites**				
6% GNP/PVDF	94.0	6.0	-	6:0
6% CNT/PVDF	94.0	-	6.0	0:6
**Hybrid filler composites**				
4.5% GNP/1.5% CNT/PVDF	94.0	4.5	1.5	4.5:1.5
3% GNP/3% CNT/PVDF	94.0	3.0	3.0	3.0:3.0

* hot-pressed PVDF powder.

**Table 3 nanomaterials-14-00901-t003:** Total crystallinity and β-phase content in PVDF and its nanocomposites.

Composition (Hot-Pressed Samples)	Aα + γ	Aα	Aβ	A_amorph._	A_α,γ,β_	Xc %	α + γ %	α %	β %
PVDF*	22,704	275	139	10,037	23,118	70.6 ± 2.3	69.3 ± 2.2	0.9 ± 0.10	0.4 ± 0.01
PVDF	20,419	517	71	11,460	21,008	64.7 ± 0.8	62.9 ± 0.8	1.6 ± 0.02	0.2 ± 0.01
6% GNP/PVDF	17,009	474	1037	14,917	18,521	55.4 ± 0.7	50.9 ± 0.6	1.4 ± 0.03	3.1 ± 0.08
4.5% GNP/1.5% CNT/PVDF	10,836	90	2617	19,710	13,543	40.6 ± 0.5	32.5 ±0.4	0.3 ± 0.01	7.8 ± 0.23
3% GNP/3% CNT/PVDF	13,088	461	3919	15,333	17,468	53.2 ± 1.0	39.9 ± 0.7	1.4 ± 0.10	11.9 ± 0.51
6% CNT/PVDF	12,575	278	4064	19,394	16,917	46.5 ± 4.5	34.6 ± 3.6	0.7 ± 0.46	11.2 ± 0.39

* hot-pressed PVDF powder.

**Table 4 nanomaterials-14-00901-t004:** Melting and crystallization temperatures and the degree of crystallinity obtained based on the DSC measurements of PVDF and its nanocomposites.

Composition	ΔHm [J/g]	T_m1_, °C	T_m2_, °C	T_c_, °C	Xc, %
First Heating Cycle					
PVDF*	53.1	169.9	-	134.5	50.7 ± 3.6
PVDF	53.2	170.1	-	139.7	50.8 ± 0.5
6% GNP–PVDF	53.4	170.4	-	141.6	54.3 ± 0.7
4.5% GNP–1.5% CNT–PVDF	46.0	170.3	-	143.1	46.7 ± 0.3
3% GNP–3% CNT–PVDF	52.6	169.7	174.8	144.2	53.4 ± 1.3
6% CNT–PVDF	53.5	170.9	175.7	144.5	53.5 ± 0.4
Second Heating Cycle					
PVDF*	51.9	169.5	-	135.7	49.6 ± 0.6
PVDF	51.7	168.0	-	139.9	49.4 ± 0.4
6% GNP–PVDF	52.0	169.1	-	141.6	52.8 ± 1.1
4.5% GNP–1.5% CNT–PVDF	48.3	169.6	-	143.2	49.1 ± 0.5
3% GNP–3% CNT–PVDF	50.9	169.1	172.9	144.2	51.7 ± 1.4
6% CNT–PVDF	50.2	169.5	173.8	144.5	51.0 ± 0.8

* hot-pressed PVDF powder.

**Table 5 nanomaterials-14-00901-t005:** Maximum temperature, current, and heat generated during the heating at applied 8 and 10 V.

Sample	Constant Voltage of 8 V (1)			Heat 1 (J)	Constant Voltage of 10 V (2)			Heat 2 (J)
	Temperature (°C)	Current (A)	Time (s)		Temperature (°C)	Current (A)	Time (s)	
6% GNP–PVDF	50.6	0.12	600	556.6	90.3	0.20	600	1170.0
4.5% GNP–1.5% CNT–PVDF	44.3	0.13	600	643.8	105.5	0.34	600	2039.0
3% GNP–3% CNT–PVDF	46.9	0.10	600	459.7	77.0	0.16	600	986.0
6% CNT–PVDF	42.5	0.08	600	368.8	66.2	0.14	600	892.6

## Data Availability

All data are contained within the article.

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
