# Peer review of "PVDF Hybrid Nanocomposites with Graphene and Carbon Nanotubes and Their Thermoresistive and Joule Heating Properties"

_nanomaterials, 2024, doi:10.3390/nano14110901_

Round 1
Reviewer 1 Report
Comments and Suggestions for Authors
The authors present "PVDF hybrid nanocomposites with graphene and carbon nanotubes and their thermoresistive and Joule heating properties." However, prior to publication, it is advisable to conduct a thorough revision of the manuscript.
1) In the abstract, all abbreviations such as PVDF must be explained.
2) Line 102: How is it determined that at 6 wt% the nanofiller percolates?
3) Do you have information regarding the Outer diameter and the Inner diameter, measured in nanometers?
4) How do the significantly different temperatures used in the extrusion process (160–175 °C) compared to those in the hot pressing method (230°C) affect the properties of the resulting nanocomposite samples?
5) What were the operating conditions for the accelerating voltage, probe current, and counting time in the SEM measurements?
6) All ICDD Database software PDF cards must be included, including that for C60.
7) Since there is no baseline subtraction and normalized intensity in the XRD patterns, how was the deconvolution performed?
8) Lines 232-238: How does the filler affect the crystallinity of the composites? How can this be explained?
9) How was Tm2 calculated in Figure 4? Did the authors conduct a deconvolution?
10) SEM microphotographs should be more descriptive, detailing how readers can discern differences between different microphotographs, and consider using arrows to highlight specific areas of interest.
11) Lines 359-366: I disagree with presenting unpublished results. We should only present results that we can confidently consider correct.
12) What is the standard error in the Temperature coefficient of resistivity (TCR) as shown in Figure 8?
13) A connection between the results needs to be established between Thermoresistive behavior and Structural and morphological characterization.
Comments on the Quality of English LanguageMinor editing of English language required
Author Response
Response to Reviewer #1 Comments
Comments and Suggestions for Authors
The authors present "PVDF hybrid nanocomposites with graphene and carbon nanotubes and their thermoresistive and Joule heating properties." However, prior to publication, it is advisable to conduct a thorough revision of the manuscript.
Authors thank to the Reviewer #1 for the valuable comments and suggestions and respond in point-by-point style. The corrections based on the Reviewer #1 comments and suggestions are in green in the revised paper.
Comments 1: In the abstract, all abbreviations such as PVDF must be explained.
Response 1: The correction is added in Line 19.
Comments 2: Line 102: How is it determined that at 6 wt% the nanofiller percolates?
Response 2: Based on the resistivity measurements percolation threshold between 2 – 3 wt.% carbon filler is observed in Lines 400-404 of the revised paper. To ensure the high conductivity needed for the thermoresistive and Joule heating measurements in the paper is worked with 6 wt.% nanocomposites.
Comments 3: Do you have information regarding the Outer diameter and the Inner diameter, measured in nanometers?
Response 3: According to the producer of Nanocyl NC 7000, the Outer diameter of multi-walled carbon nanotubes (MWCNTs) is 9.5 nm and the Inner diameter is below < 5 nm. The producer provides information regarding the average size, D50 < 50 μm of the graphene nanoplatelets (GNPs) because GNPs are shaped in layers due to their two-dimensional structure. This information is provided in Table 1, in Line 120.
Comments 4: How do the significantly different temperatures used in the extrusion process (160–175 °C) compared to those in the hot-pressing method (230°C) affect the properties of the resulting nanocomposite samples?
Response 4: The following text, marked in green is added:
In Lines 132-134:
"The temperatures used for the extrusion process (160–175 °C) in the range of the melting point of the PVDF aimed the good integration of the nanoparticles inside the polymer matrix."
And in Lines 144-146:
"to promote the β phase formation [34]. This elevated temperature was also applied in order to obtain thin test samples with a thickness of around 650 μm."
Comments 5: What were the operating conditions for the accelerating voltage, probe current, and counting time in the SEM measurements?
Response 5: The following conditions were mentioned in Lines 167-168:
"The SEM images were taken at 15kV accelerating voltage and 110mA emission current conditions." The counting time is not considered as we did not perform SEM-EDS analysis.
Comments 6: All ICDD Database software PDF cards must be included, including that for C60.
Response 6: The ICDD PDF reference marked in green was included in Line 216.
Comments 7: Since there is no baseline subtraction and normalized intensity in the XRD patterns, how was the deconvolution performed?
Response 7: The following addition to the text of the revised paper is made: in Lines: 236-242, as marked in green:
"The deconvolution was carried out by fixing the baseline parameters for each sample, allowing for an unbiased comparison of the results. The initial positions of the crystalline peaks were fixed at known values, and the amorphous halo was positioned between 17–19° and 23–25° 2θ. The area and full width at half maximum of the peaks were recalculated during the convergent iterations. The peak deconvolution method was performed using the First Derivative Peak Finding Algorithm with Origin software."
Comments 8: Lines 232-238: How does the filler affect the crystallinity of the composites? How can this be explained?
Response 8: The presence of carbon-based nanofillers had been proven to enhance the β crystal phase formation in PVDF. This could be seen in the XRD results in the Table 3 (Line 247), as well as in the DSC thermograms of cooling in Fig. 4b (Line 286).
Comments 9: How was Tm2 calculated in Figure 4? Did the authors conduct a deconvolution?
Response 9: The melting temperatures Tm1 and Tm2 were analyzed using TA Universal Analysis Software, TA Instruments with the function Peak Maximum which determined the position of the melting peaks.
Comments 10: SEM microphotographs should be more descriptive, detailing how readers can discern differences between different microphotographs, and consider using arrows to highlight specific areas of interest.
Response 10: Figure 5 is corrected and the following addition to the text, marked in green is proposed in Lines 335-341:
"In Figure 5b2 for the composition of 6%CNT/PVDF many circle-shaped inclusions are embedded in the structure which indicates good affinity between the MWCNTs and PVDF indicating the coverage of the nanoparticle. For the composition of 6%GNP/PVDF and 4.5%GNP/1.5%CNT/PVDF similar cohesiveness was found. In all the nanocomposites in Figure 5b–e, it is observed that the nanofiller particles on the cut surfaces are fully coated with the matrix polymer, meaning that the fracture was done through the polymer matrix and not on the polymer-filler interfaces."
Comments 11: Lines 359-366: I disagree with presenting unpublished results. We should only present results that we can confidently consider correct.
Response 11: We accept the reviewer’s comment and have made the following changes in the revised paper. The text for unpublished results in Lines 400-404 was deleted and replaced by new text marked in green:
"Our measurements show the resistivity data for the MWCTN/PVDF nanocomposites of 3.9×106 Ω⋅m for 1.5%CNT/PVDF; 1.1×106 Ω⋅m for 2%CNT/PVDF; and 2.0 Ω⋅m for 3%CNT/PVDF. Therefore, based on these results, the percolation threshold of the MWCNT/PVDF nanocomposites appears between 2 wt.% and 3 wt.% MWCNTs."
Comments 12: What is the standard error in the Temperature coefficient of resistivity (TCR) as shown in Figure 8?
Response 12: The standard error of TCR was determined for all compositions studied and included in the corrected Figure 8, in Line 440.
Comments 13: A connection between the results needs to be established between Thermoresistive behavior and Structural and morphological characterization.
Response 13: A connection between the results of thermoresistive behavior and structural characterization has been established in the paper, and this may be seen in the following paragraphs, where the message we want to convey as authors is the following:
In Lines 421-427: The Thermoresistive behavior of 4.5%GNP/ 1.5%CNT/PVDF has been connected to the lower crystallinity, which could facilitate the thermal expansion of the PVDF matrix.
In Lines 392-394: The temperature influence on the resistivity has been expressed in changes of the percolation network of hybrid nanocomposites.
In Lines 462-464: The Joule heating effect in the 4.5%GNP/1.5%CNT/PVDF nanocomposite has been correlated to the disruption of the percolated structure of carbon nanofillers by the thermal expansion of the polymer.
Reviewer 2 Report
Comments and Suggestions for Authors
In this paper, the authors prepared nanocomposites of PVDF with carbon nanotubes (CNT) or graphene (GN) or a combination of these two fillers., through melt extrusion. They characterized the materials with XRD, DSC SEM and evaluated the materials for their thermo-resistivity and Joule heating performance. In general, this is a well-written paper. The experimental part is well- presented and the introduction is sufficiently informative and relative to the topic of interest. The results are presented in a clear manner and the conclusions are supported by the presented results. Prior to publication the following issues can be addressed:
1. Lines 61-62. In the abstract you mention that the fillers in your case resulted in less crystallinity and more amorphous phase. Any explanation for your disagreement with the literature? The literature that you mention refer to α, β or what phase?
2. Line 78. Please briefly explain what is the self-regulating effect.
3. XRD and DSC results. According to XRD the nanocomposites exhibit quite lower overall crystallinity but from the heat of fusion values this does not seem to be the case since all samples exhibit pretty much the same heat of fusion. Please provide an explanation for this inconsistency between the XRD and DSC results regarding the degree of crystallinity.
4. lines 354-357. Is any thermal effect detected around 75 oC in the DSC curves? It is likely that these changes in the percolation network that you mentioned are detectable in DSC. I think it would be helpful to present the DSC at around 75 oC.
5. Please compare your results with the ones of the literature regarding the thermo-resistivity and joule heating values.
Author Response
Response to Reviewer #2 Comments
Comments and Suggestions for Authors
In this paper, the authors prepared nanocomposites of PVDF with carbon nanotubes (CNT) or graphene (GN) or a combination of these two fillers., through melt extrusion. They characterized the materials with XRD, DSC SEM and evaluated the materials for their thermo-resistivity and Joule heating performance. In general, this is a well-written paper. The experimental part is well- presented and the introduction is sufficiently informative and relative to the topic of interest. The results are presented in a clear manner and the conclusions are supported by the presented results. Prior to publication the following issues can be addressed.
Authors would like to thank to the Reviewer #2 comments and suggestions and respond in point-by-point style. The changes in the revised paper based on the Reviewer # 2 comments are in yellow.
Comments 1: Lines 61-62. In the abstract you mention that the fillers in your case resulted in less crystallinity and more amorphous phase. Any explanation for your disagreement with the literature? The literature that you mention refer to α, β or what phase?
Response 1: This observation was noticed in Figure 3 (in Line 243). Initially, the enlargement of the amorphous halo was observed by comparing hot-pressed PVDF powder to the extruded and hot-pressed samples, which led to a lower crystalline fraction. The two-step process of extrusion and hot-pressing decreases the crystalline fraction, as seen in Figure 3 and Table 3 (in Lines 243 and 247). On the other side, the addition of carbon nanofillers leads to an increase in the β phase. In the literature, it is referred to as the characteristic peaks found for the α, β, and γ phases. Therefore, we can conclude for our results that two phenomena are taking place simultaneously: the process inhibits the crystallization while the carbon-based nanofiller enhance the β phase formation.
Comments 2: Line 78. Please briefly explain what is the self-regulating effect.
Response 2: Added in Lines 78-79: "(an equilibrium of the generated temperature reached at applied voltage)"
Comments 3: XRD and DSC results. According to XRD the nanocomposites exhibit quite lower overall crystallinity but from the heat of fusion values this does not seem to be the case since all samples exhibit pretty much the same heat of fusion. Please provide an explanation for this inconsistency between the XRD and DSC results regarding the degree of crystallinity.
Response 3: The XRD analysis is performed for a limited range of 15° and 30° 2θ, and it is used to perform the qualitative analysis of the α, β, and γ phases in the PVDF and its nanocomposites. While DSC analysis allows determining mainly the total crystallinity by using the enthalpy of melting.
The following addition to the text of the revised paper is made: in Lines: 206-209, as marked in yellow.
"The XRD analysis was performed for limited range of the diffractogram where the characteristic peaks for the α, β and γ phases were observed and it was used to perform the qualitative analysis for the information of the phases in the PVDF and its nanocomposites."
Comments 4: lines 354-357. Is any thermal effect detected around 75 °C in the DSC curves? It is likely that these changes in the percolation network that you mentioned are detectable in DSC. I think it would be helpful to present the DSC at around 75 °C.
Response 4: The following text, marked in yellow is added:
In Lines 391-392:
"The DSC curve did not show a thermal effect around 75 °C."
And in Lines 394-397:
"The observed changes in the percolation network around 75 °C referred to disruption of the percolation network due to the thermal expansion of the matrix which happened at mesoscale which could not be detected by the DSC measurement."
Comments 5: Please compare your results with the ones of the literature regarding the thermo-resistivity and joule heating values.
Response 5: As shown in the Introduction part of the paper, in the referred literature we have not found articles reporting the thermoresistivity and Joule heating results for PVDF nanocomposites incorporating GNPs, MWCNTs and their hybrid combinations, in order to compare our results with the ones of the literature. Our results are original and novel, where the thermoresistivity and Joule heating are related to the percolation structure of nanofillers and the crystal structure of the PVDF nanocomposites.
Round 2
Reviewer 1 Report
Comments and Suggestions for Authors
Accept in present form